# *Ustilago maydis* Trf2 ensures genome stability by antagonizing Blm-mediated telomere recombination: Fine-tuning DNA repair factor activity at telomeres through opposing regulations

**Shahrez Syed**[1], **Sarah Aloe**[1], **Jeanette H. Sutherland**[1], **William K. Holloman**[1], **Neal F. Lue**[1,2]*

1 Department of Microbiology & Immunology, W. R. Hearst Microbiology Research Center, Weill Cornell Medicine, New York, New York, United States of America, 2 Sandra and Edward Meyer Cancer Center, Weill Cornell Medicine, New York, New York, United States of America

* nflue@med.cornell.edu

**Data Availability Statement:** All relevant data are within the manuscript and its Supporting Information files.

## Abstract

TRF2 is an essential and conserved double-strand telomere binding protein that stabilizes chromosome ends by suppressing DNA damage response and aberrant DNA repair. Herein we investigated the mechanisms and functions of the Trf2 ortholog in the basidiomycete fungus *Ustilago maydis*, which manifests strong resemblances to metazoans with regards to the telomere and DNA repair machinery. We showed that *Um*Trf2 binds to Blm *in vitro* and inhibits Blm-mediated unwinding of telomeric DNA substrates. Consistent with a similar inhibitory activity *in vivo*, over-expression of Trf2 induces telomere shortening, just like deletion of *blm*, which is required for efficient telomere replication. While the loss of Trf2 engenders growth arrest and multiple telomere aberrations, these defects are fully suppressed by the concurrent deletion of *blm* or *mre11* (but not other DNA repair factors). Over-expression of Blm alone triggers aberrant telomere recombination and the accumulation of aberrant telomere structures, which are blocked by concurrent Trf2 over-expression. Together, these findings highlight the suppression of Blm as a key protective mechanism of Trf2. Notably, *U. maydis* harbors another double-strand telomere-binding protein (Tay1), which promotes Blm activity to ensure efficient replication. We found that deletion of *tay1* partially suppresses the telomere aberration of Trf2-depleted cells. Our results thus point to opposing regulation of Blm helicase by telomere proteins as a strategy for optimizing both telomere maintenance and protection. We also show that aberrant transcription of both telomere G- and C-strand is a recurrent phenotype of telomere mutants, underscoring another potential similarity between double strand breaks and de-protected telomeres.

**Funding:** This work was supported by National Science Foundation Grants MCB-1817331 and MCB-2246561 (URL: https://www.nsf.gov/) and National Institutes of Health Grant GM107287 (URL: https://www.nih.gov/).The funders played no role in the study design, data collection and analysis, decision to publish, or preparation of the manuscript.

**Competing interests:** The authors have declared that no competing interests exist.

## Author summary

The ends of linear chromosomes are protected from abnormal repair by a collection of telomere proteins. One protein that plays an especially prominent role is TRF2, which binds to double-stranded telomere repeats. In this study, we analyzed the mechanisms and functions of Trf2 in a yeast-like fungus named *Ustilago maydis*, which manifests a high degree of similarity to animal cells with respect to telomere regulation. We showed that Trf2 binds directly to a conserved DNA helicase called Blm and inhibits the ability of Blm to unwind telomeric DNA in a purified, cell-free reaction. We also used over-expression and depletion of either Trf2 or Blm or both to demonstrate an inhibitory effect of Trf2 on Blm function *in vivo*. For example, depletion of Trf2 triggers Blm-dependent telomere aberrations and cell death. Interestingly, another double-strand telomere binding protein named Tay1 was found to stimulate Blm activity to promote telomere replication. Together, our results indicate that *U. maydis* optimizes Blm function through opposing regulation of its activity via distinct telomere proteins. We also detected high levels of abnormal transcripts that correspond to both strands of telomeres in a variety of telomere mutants, suggesting that de-protected telomeres are permissive substrates for the transcription apparatus.

## Introduction

Telomeres promote genome integrity by stabilizing chromosome ends against aberrant rearrangements [1,2]. Stabilization requires the formation of telomere-specific nucleoprotein complexes, which prevent chromosome ends from being recognized as double strand breaks (DSBs) and from instigating DNA damage response (DDR) or abnormal repair [3,4]. This concept stipulates that telomeres must directly or indirectly suppress factors in the DDR and DNA repair pathways. However, this relationship is complicated by the fact that DNA repair factors are themselves important for the maintenance of telomere DNA. Owing to the propensity of G-rich telomere DNAs to form replication barriers such as G-quadruplexes and T-loops, efficient telomere replication requires the participation of DNA repair factors such as Rad51 and Blm to stabilize stalled forks and unwind replication obstacles [5,6]. Therefore, instead of unconditionally inhibiting DNA repair, telomere proteins must also permit DNA repair factors to act in the appropriate setting to promote telomere maintenance and safeguard genome stability. Understanding the complicated and dichotomous relationships between telomeres and the DNA repair machinery is a central task in telomere research.

The functions of telomeres in mammalian cells are primarily mediated by the shelterin complex, a six-protein assembly that coats double-stranded telomere DNA repeats as well as the terminal 3' overhang (G-overhang) [3]. Among the components of shelterin are two proteins that directly recognize double-stranded telomere repeats (TRF1 and TRF2), one that recognizes the G-strand overhang (POT1), and several associated and bridging factors (RAP1, TIN2, and TPP1). Interestingly, the two double-strand telomere binding proteins, TRF1 and TRF2, evidently arose through gene duplication during metazoan evolution and mediate quite distinct functions in telomere regulation [3,7]. TRF1 is crucial for telomere DNA maintenance; it facilitates efficient telomere replication by recruiting or activating the BLM helicase to unwind replication barriers [8]. In mouse cells, TRF1 also suppresses recombination and BIR through as yet undefined mechanisms [9,10]. By contrast, TRF2 serves essential functions in telomere protection; loss of TRF2 function results in ATM activation and chromosome fusions

due to classical non-homologous end joining (cNHEJ), indicating that these DDR and repair systems are the main targets of TRF2 inhibition [11]. The inhibition of cNHEJ by TRF2 is achieved through multiple mechanisms including TRF2-mediated T-loop formation and suppression of 53BP1 accumulation [12,13]. Also potentially playing a role in the inhibition is the binding of TRF2 to KU70, an essential cNHEJ factor [14]. It should be noted, however, that the mechanisms and functions of TRF2 are far from fully understood. For example, overexpression of TRF2 has been shown to cause accelerated telomere shortening as well as telomere anaphase bridges, which lacks clear mechanistic explanation. In addition, TRF2 has been shown to harbor other activities that may be relevant to telomere regulation [15,16]. Indeed, multiple studies have indicated that TRF2 can bind directly to the BLM helicase [17,18], which is implicated in promoting telomere replication and telomere recombination (e.g., in the alternative lengthening of telomeres, or ALT pathway) [8,19,20]. However, the *in vivo* significance of TRF2-BLM interaction remains unclear. Understanding the full set of interaction partners for TRF2 and the functional consequences of these interactions is necessary for delineating its telomere regulatory mechanisms.

The budding yeast *Saccharomyces cerevisiae* and fission yeast *Schizosaccharomyces pombe* have been used extensively as model systems to investigate telomere regulation. However, both models manifest substantial differences from mammalian telomeres, including the identity and mechanisms of duplex telomere binding proteins [21]. As a complementary experimental system, we have examined the basidiomycete fungus *Ustilago maydis*, originally developed by Robin Holliday to investigate recombinational repair [22]. In comparison to budding and fission yeasts, *U. maydis* bears greater resemblances to mammalian cells with respect to the telomere and DNA repair machinery [23,24]. *U. maydis* also harbors the canonical 6-bp telomere repeat sequence ($[TTAGGG]_n/[CCCTAA]_n$), which recapitulates the physical-chemical properties of mammalian telomeric DNA. The greater similarity of *U. maydis* to mammals can be explained by the more basal position of basidiomycetes in fungal phylogeny, which accounts for their closer evolutionary kinship to metazoans. In previous studies, we identified two duplex telomere repeat-binding factors in *U. maydis*, named Tay1 and Trf2. These proteins are structurally distinct and mediate non-overlapping functions in telomere maintenance and protection [25,26]. Trf2 is structurally similar to metazoan TRFs and is critical for telomere protection; loss of Trf2 triggers growth arrest and a multiplicity of telomere defects [26]. Tay1, by contrast, is a fungi-specific telomere protein that promotes telomere replication and recombination by enhancing Blm activity [26,27]. Thus, even though Tay1 and mammalian TRF1 do not share the same molecular ancestry, they have apparently acquired the same function, thus underscoring the evolutionary advantage of enhancing Blm activity so as to ensure efficient telomere replication.

Herein we report our further investigation of *U. maydis* Trf2, the essential duplex telomere-binding protein that is orthologous to mammalian TRF2. We showed that the telomere deprotection phenotypes of *trf2*-deficienct cells can be greatly suppressed by deleting *blm* or *mre11*, but not several other DNA repair factors. Trf2 inhibits the unwinding of telomeric DNA by Blm *in vitro* and overexpression of Trf2 *in vivo* induces telomere shortening just like deletion of *blm*. Moreover, while the over-expression of Blm triggers telomere recombination, this phenotype can be suppressed by concurrent overexpression of Trf2. Together with previous findings, our results point to opposing regulation of Blm helicase by telomere proteins as a strategy for fine-tuning Blm function–such that this helicase can promote telomere maintenance without triggering telomere de-protection. Previous findings on mammalian TRF1, TRF2 and BLM suggest that this regulatory strategy may be conserved in evolution, with TRF1 and TRF2 acting in analogous manners as *U. maydis* Tay1 and Trf2.

## Results

### The growth and telomere defects triggered by Trf2 deficiency are suppressed by deletions of *blm* and *mre11*

Because *U. maydis trf2* is essential, we assessed the consequences of Trf2 loss using a conditional mutant (*trf2^crg1^*) in which *trf2* expression is controlled by the arabinose-dependent *crg1* promoter. In this mutant, *trf2* RNA levels were substantially elevated in YPA and repressed in YPD in comparison to the parental control (UCM) (Fig 1A). We showed previously that Trf2 deficiency (*trf2^crg1^* grown in YPD) results in growth arrest and a constellation of telomere

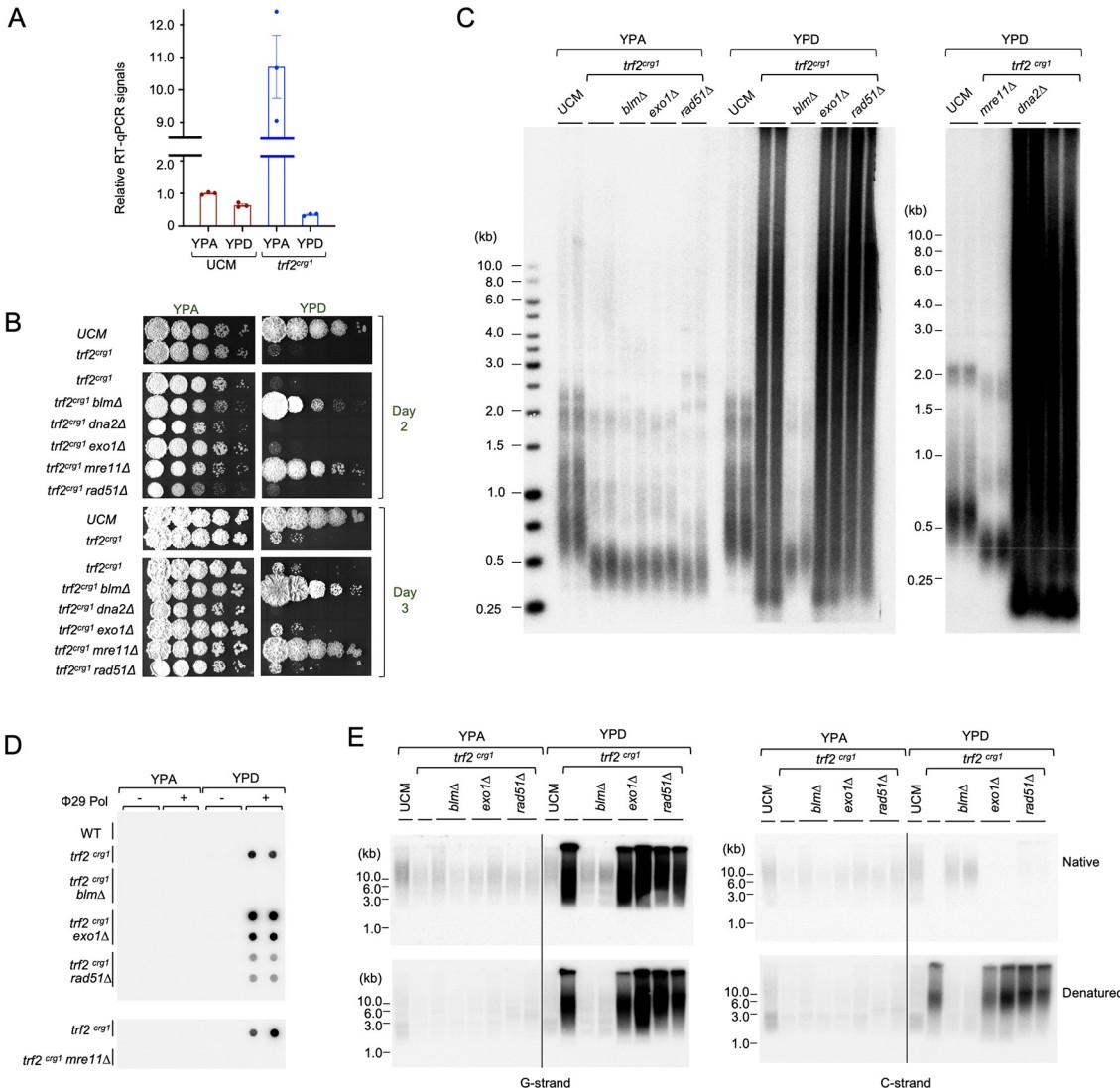

**Fig 1. The roles of specific DNA repair factors in causing growth defects and telomere aberrations in Trf2-deficient U. maydis cells. (A)** RT-qPCR analysis of *trf2* RNA levels in the UCM350 (parental) and *trf2^crg1^* strains grown in YPA or YPD. The signals were normalized to that of UCM350 grown in YPA and plotted. Data (mean ± SEM) are from three or more independent measurements. **(B)** Serial dilutions of the indicated strains were spotted onto YPA and YPD media and incubated at 30°C. Following 2 and 3 days of growth, the plates were imaged. **(C)** Genomic DNAs were isolated from the indicated strains grown in YPA or YPD, digested with PstI, and subjected to TRF Southern analysis. **(D)** Genomic DNAs from the indicated strains grown in YPA or YPD were subjected to C-circle analysis. **(E)** Genomic DNAs were isolated from the indicated strains grown in YPA or YPD, digested with PstI, and subjected to in-gel hybridization to assess the levels of G-strand ssDNA (top left panel) or C-strand ssDNA (top right panel). The gels were then denatured and re-hybridized with the same probes to determine the overall telomere content (bottom panels).

defects including telomere length heterogeneity, accumulation of telomere ssDNA, and high levels of C-circles [26]. To identify the DNA repair factors responsible for triggering the growth defect and telomere aberrations, we constructed and analyzed a series of double mutants that carry both the *trf2^{crg1}* allele and a deletion in a DNA repair gene (*blm*, *dna2*, *exo1*, *rad51* and *mre11*). When grown in YPA (*trf2* on), none of the double mutants exhibited significant growth defects (Fig 1A). However, upon transcriptional repression of *trf2^{crg1}* in YPD, all of the double mutants except *trf2^{crg1} blmΔ* and *trf2^{crg1} mre11Δ* manifested growth arrest. Telomere restriction fragment Southern analysis revealed a tight correlation between the suppression of growth defect and maintenance of normal telomere restriction fragment patterns in the double mutants, indicating that *blmΔ* and *mre11Δ* restore viability by suppressing telomere aberrations (Fig 1B). This is also consistent with the levels of G-strand ssDNA and C-circles in the double mutants; these aberrant structures were present at low or undetectable levels in *trf2^{crg1} blmΔ* and *trf2^{crg1} mre11Δ*, but at high levels in other *trf2^{crg1}* double mutants grown in YPD (Fig 1C and 1D). Together, these results indicate that Blm and Mre11 are both essential for inducing telomere aberrations in the context of Trf2-deficiency and that *U. maydis* Trf2 mediates telomere protection by antagonizing the activities of these DNA repair factors. It is worth noting that in an earlier study, we reported increased C-strand ssDNA in Trf2-depleted cells [26]. However, subsequent analysis consistently revealed high levels of G-strand ssDNA rather than C-strand ssDNA, indicating that the G-strand accumulation is the reproducible phenotype of this mutant.

Interestingly, while the *trf2^{crg1} blmΔ* and *trf2^{crg1} mre11Δ* mutants are viable on YPD, they manifested reduced growth in comparison to the wild type strain as evidenced by the mutants' small colony size and decreased growth rate (S1 Fig). The basis for this growth defect is unclear given the lack of telomere aberrations. Importantly, the defect does not appear to be the consequence of telomere shortening given that these mutants harbor similarly short telomeres in YPA and YPD and yet exhibit poorer growth in YPD (Fig 1A). Possibly even without drastic telomere aberration, the lack of Trf2 on telomeric chromatin can induce DNA-damage signaling that impairs cell proliferation.

## Trf2 overexpression induces telomere shortening

While neither *trf2^{crg1}* nor any of the double mutant exhibited growth defect when grown in YPA, they all harbor dramatically shortened but stably maintained telomeres (Fig 1B). This result suggests that Trf2 overexpression (*trf2^{crg1}* grown in YPA) induces telomere shortening. To test the generality of this observation, we constructed and analyzed another set of *trf2^{crg1}* mutants in the FB1 strain background–different from the UCM350 strain used for the majority of this study. Again, these mutants exhibited dramatically shortened telomeres in YPA and manifested growth arrest as well as telomere aberrations in YPD (S2 Fig).

We reasoned that if Trf2 overexpression in YPA triggers telomere shortening, then reducing *trf2* transcription should ameliorate the effects. To titrate the expression of Trf2, we grew *trf2^{crg1}* in 1.5% arabinose and increasing concentrations of glucose. (Previous analysis indicates that even in the presence of arabinose, glucose can inhibit *crg1* promoter activity [28].) RT-qPCR analysis confirmed the progressive reduction in *trf2* RNA levels as the glucose concentrations in the media became elevated (S3A Fig). As expected, this reduction in *trf2* RNA level is accompanied by a progressive increase in telomere lengths. At 0.125% glucose, the TRF lengths of the *trf2^{crg1}* mutant resembled those of the parental strain, whereas at lower glucose concentrations, the TRF lengths of the mutant were significantly shorter (S3B Fig).

Notably, the telomere shortening phenotype of *trf2^{crg1}* grown in YPA (henceforth designated *trf2^{O/E}*) is reminiscent of several previously characterized DNA repair mutants (e.g.,

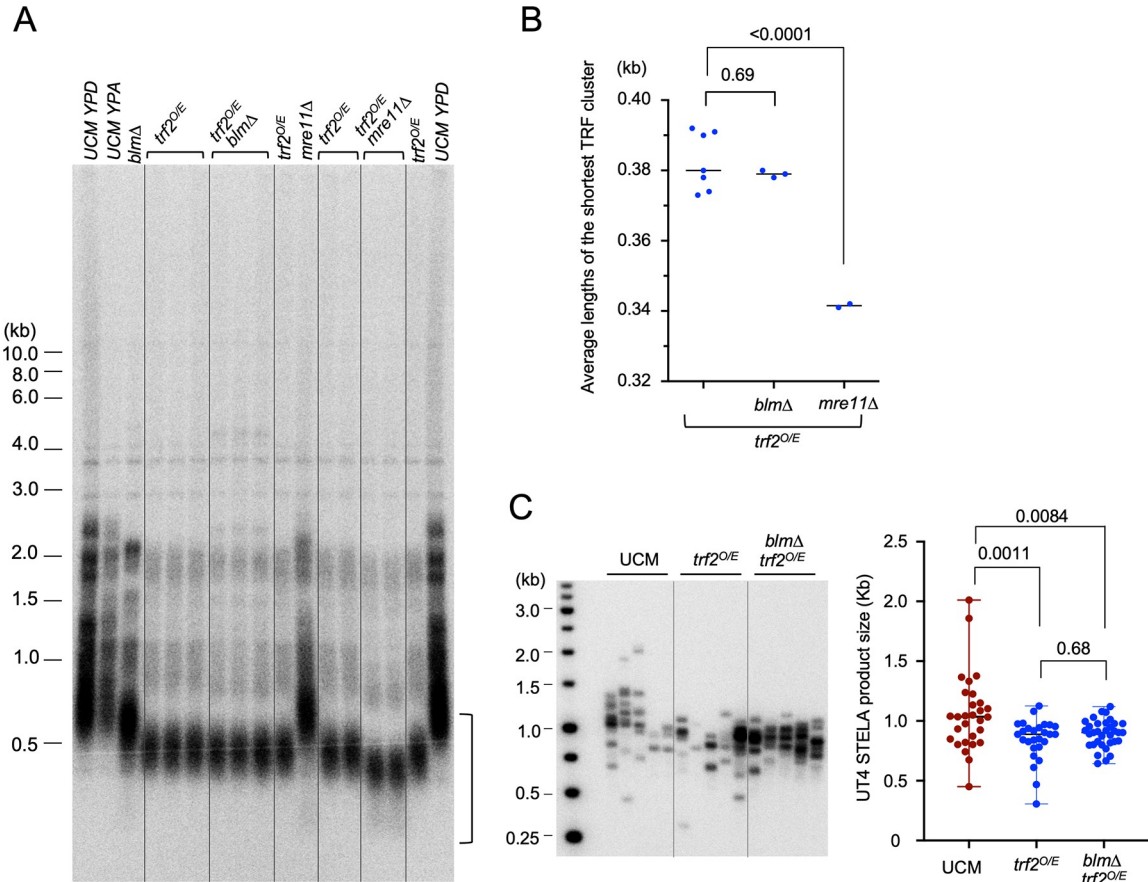

**Fig 2. Epistasis analysis of telomere shortening induced by *trf2^O/E^*, *blmΔ* and *mre11Δ*.** (A) Genomic DNAs were isolated from the indicated strains, digested with PstI, and subjected to TRF Southern analysis. The region used for determination of TRF lengths is marked by a vertical bracket. **(B)** Average TRF lengths were determined from multiple independent DNA samples from the indicated strains and plotted (all data points and the means). For this analysis, we included only signals below 0.6 kb (see the vertical bracket in Fig 2A) in order to obtain a more accurate assessment of the shortest TRF cluster. The samples were compared using ANOVA with Dunnett's multiple comparisons test and the p values displayed. **(C)** STELA analysis of telomere length distributions in UCM350, *trf2^O/E^*, and *trf2^O/E^ blmΔ* strains grown in YPA. The blot is shown on the left and the length distribution of STELA fragments plotted on the right (all data points plus the median and range). The samples were compared using ANOVA with Dunnett's multiple comparisons test and the p values displayed.

*blmΔ* and *rad51Δ*), suggesting a defect in telomere replication [25,29]. Given the apparent ability of Trf2 to antagonize Blm and Mre11 activity, a plausible explanation is that Trf2 overexpression may compromise telomere maintenance by inhibiting the activity of these two factors, which had previously been implicated in efficient telomere replication [30,31]. To test this hypothesis, we examined the epistasis relationship between the telomere shortening triggered by *trf2^O/E^* and by deletions of *blm* and *mre11* (Fig 2). The results indicate that *trf2^O/E^* is epistatic to *blmΔ* but not to *mre11Δ*. While *trf2^O/E^* alone manifests more severe telomere shortening than *blmΔ*, the *trf2^O/E^ blmΔ* double mutant harbors similar telomeres as *trf2^O/E^* (Fig 2A and 2B). By contrast, *trf2^O/E^ mre11Δ* exhibited shorter telomeres than either *trf2^O/E^* or *mre11Δ* cells. We further examined telomere length distributions in the *trf2^O/E^* and *trf2^O/E^ blmΔ* mutants using STELA and again found no significant differences (Fig 2C) [31]. Additional analysis suggests that with regard to telomere shortening, *trf2^O/E^* is also epistatic to *exo1Δ* (S4A Fig), but not to *dna2Δ* (S4B Fig). Together, these results underscore the potential of Trf2 to antagonize DNA repair factors and highlight an especially interesting interaction between Trf2

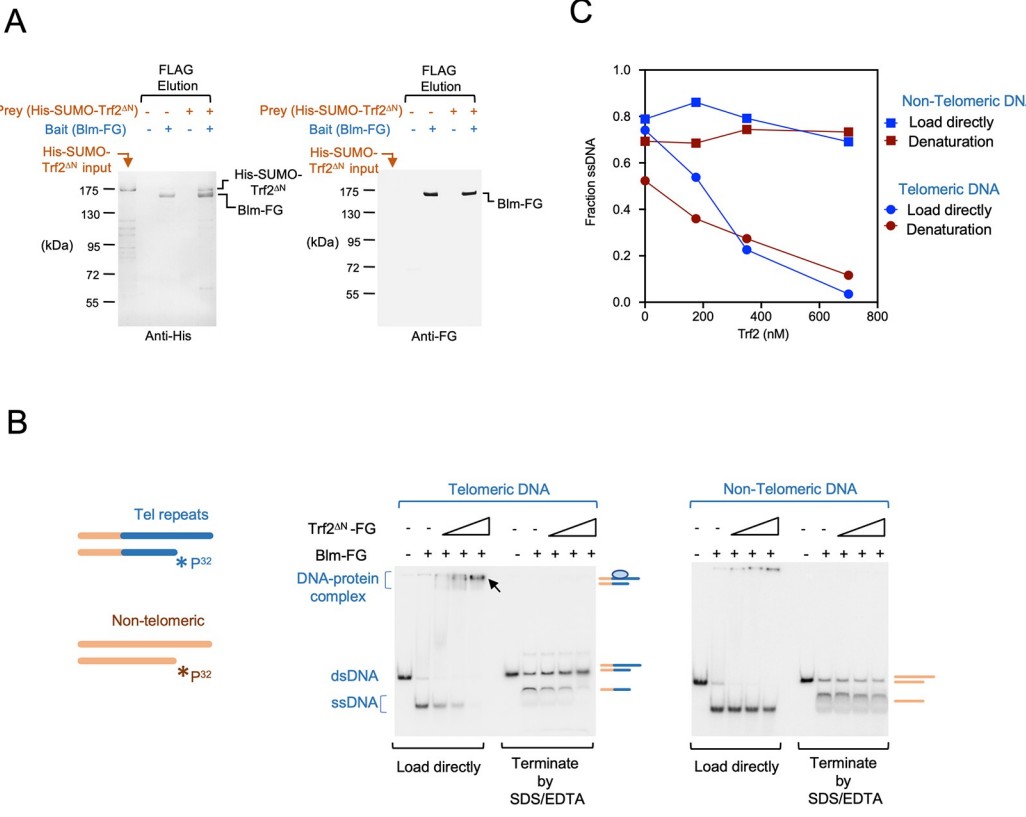

**Fig 3. Blm-Trf2 interaction *in vitro* and the effects of Trf2 on Blm helicase activity. (A)** The elution samples from a Blm-FG/His-SUMO-Trf2$^{\Delta N}$ pull down experiment were analyzed by Western using anti-His$_6$ (left panel) and anti-FLAG (right panel) antibodies, respectively. For comparison, the His-SUMO-Trf2$^{\Delta N}$ input fraction in the pull-down assays were also analyzed (leftmost lane in each panel). Note that Blm-FG cross-reacted with the anti-His antibodies; the bands labeled as Blm-FG in this blot are only found in samples containing this protein and their mobility is identical to that of Blm-FG in the anti-FG blot. **(B)** The effects of Trf2$^{\Delta N}$ (175 nM, 350 nM and 700 nM) on Blm helicase activity were analyzed using substrates with and without telomeric repeats (left and right panels, respectively). Following incubation, the reaction products were analyzed by electrophoresis with or without prior treatment with SDS/EDTA (to dissociate protein-DNA complex). The bands corresponding to the duplex DNA substrates, ssDNA products and protein-DNA complexes are marked by schematic illustrations to the right of the panels. **(C)** The fraction of radioactivity in the form of ssDNA in each sample was determined (as ssDNA signal/total signal) in each lane and plotted. The experiment was repeated three times using different Trf2 concentrations, and the inhibitory effect of Trf2 on ssDNA accumulation (telomeric DNA assays) was consistently observed.

and Blm. For the remainder of this study, we focus on the regulation of Blm by telomere proteins.

## Trf2 binds to Blm and inhibits Blm-mediated unwinding of telomeric DNA substrates *in vitro*

The foregoing results suggest that a main function of *Um*Trf2 is to restrain Blm activity at telomeres. We therefore investigated Trf2-Blm interactions as well as the effects of Trf2 on the Blm helicase activity *in vitro*. These assays were performed using an N-terminally truncated Trf2 (Trf2$^{\Delta N}$) because the full length protein could not be expressed in *E. coli* [26]. Pull down assays using recombinant Blm-FG (Blm with a C-terminal FLAG tag) and His-Trf2$^{\Delta N}$ (Trf2$^{\Delta N}$ with an N-terminal His$_6$-SUMO tag) revealed a modest but detectable interaction between these two proteins (Fig 3A). To assess the effect of Trf2 on Blm helicase activity, we utilized two 3'-tailed substrates, one with completely non-telomeric sequences and the other with

telomere repeats in the duplex region and the 3' tail (Fig 3B). The substrates were 5' end-labeled on the recessed strands, incubated with Blm and ATP as well as increasing concentrations of Trf2, and then analyzed directly by electrophoresis. Alternatively, the reactions were first terminated with the addition of EDTA and SDS, and then subjected to electrophoresis. In either protocol, the unwinding of substrates by Blm should result in the conversion of the labeled, duplex DNA species into a faster migrating single-stranded DNA band. As shown in Fig 3B and 3C, Trf2$^{\Delta N}$ inhibited Blm unwinding of telomeric DNA, but had no effect on non-telomeric DNA substrates. At 700 nM Trf2$^{\Delta N}$, the unpairing of telomeric DNA duplex by Blm was almost completely inhibited, whereas that of non-telomeric DNA was unaffected. This result suggests that the DNA-binding activity of Trf2$^{\Delta N}$ is important for Blm inhibition. Indeed, the loss of ssDNA products in telomeric DNA reactions that contained high concentrations of Trf2 is accompanied by the appearance of slowly migrating DNA that likely represents Trf2-DNA complexes (Fig 3B, marked by an arrow). A low level of DNA-protein complex was also detected in the non-telomeric DNA reactions, which may be due to the non-specific DNA-binding activity of Trf2 [32]. Together, these results indicate that Trf2 has the capacity to inhibit Blm-mediated unwinding of telomeric DNA. Whether the modest physical interaction between Trf2 and Blm plays a role in this inhibition is unclear.

## Overexpression of Blm triggers telomere aberrations

The findings described thus far suggest that stringent control of Blm activity by Trf2 is required for telomere protection. This in turn raises the possibility that overexpression of Blm could override the control and trigger telomere de-protection. We tested this possibility by analyzing conditional *blm* mutants (*blm$^{crg1}$*) that overexpress *blm* in YPA. As predicted, the *blm* RNA levels in *blm$^{crg1}$* clones were substantially elevated when the cells are grown in YPA (~ 7-fold higher than the UCM350 control), but greatly repressed (~ 1/5 of the control) when the cells are grown in YPD (Fig 4A). Notably, overexpression of Blm is associated with significant growth deficiency; in comparison to the *UCM350* and *blmΔ* strains, the *blm$^{crg1}$* mutants exhibited reduced growth on YPA, but not on YPD (Fig 4B). Analysis of telomere lengths and structures in the Blm-overexpressing clones revealed multiple aberrations, including telomere length heterogeneity and accumulation of extra-short telomeric fragments (Fig 4C, marked by a bracket), as well as high levels of C-circles (Fig 4D). The extra-short telomeric fragments are extra-chromosomal because they are detectable in the absence of restriction enzyme cleavage (S5A Fig). All of these telomere aberrations exhibited similar kinetics following Blm overexpression in YPA: they were mostly absent at the 4 h timepoint, reached near maximal severity following 24 h of growth in YPA, and were maintained at similar levels up to the 72 h timepoint (Fig 4C and 4D). The defects were also reversible upon repression of Blm in YPD (S5B Fig). Analysis of telomere ssDNA revealed a more complex pattern: the C-strand ssDNA exhibited a large increase at 24 h followed by moderate decline, whereas the G-strand ssDNA exhibited a persistent reduction (Fig 4E and 4F). (For the plots of ssDNA dynamics in *blm$^{crg1}$* clones shown in Fig 4F, we used the 4 h timepoint as the starting reference since aberrant telomere structures were mostly undetectable at this timepoint.) We conclude that abnormally high levels of Blm is sufficient to trigger telomere de-protection, resulting in a constellation of telomere abnormalities. Notably, the telomere phenotypes of *blm$^{crg1}$* grown in YPA share similarities with but are not identical to those of Trf2-deficient cells. For example, the abnormal telomere fragments in the Blm-overexpressing cells were predominantly short, whereas the Trf2-deficient cells accumulate high levels of both long and short telomeric fragments (compare Figs 4C and 1B). In addition, the former exhibits high levels of C-strand ssDNA, whereas the latter high levels of G-strand ssDNA. These results suggest that Blm is not the only target

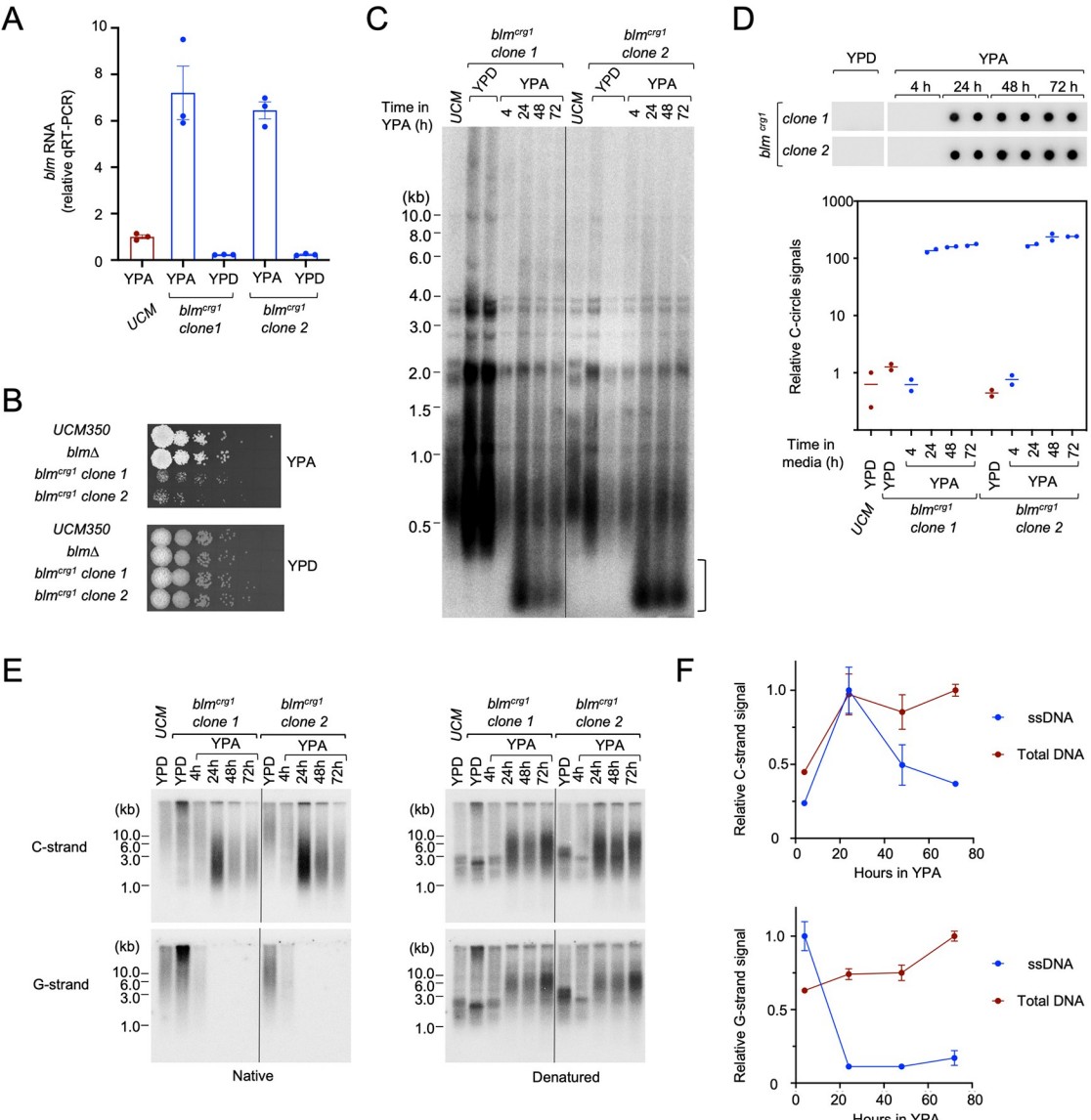

**Fig 4. Blm overexpression triggers growth defects and telomere deprotection. (A)** RT-qPCR analysis of *blm* RNA levels in the parental strain and *blm^crg1* mutants. Data (mean ± SEM) were from three independent measurements. **(B)** Serial dilutions of the indicated *U. maydis* strains were spotted onto YPA or YPD plates and incubated at 30°C. The plates were imaged following 2 days of growth. **(C)** Chromosomal DNAs from the indicated strains grown in the specified media were isolated and subjected to TRF Southern analysis. The *blm^crg1* clones were cultured in YPA for different durations prior to DNA isolation and Southern analysis. **(D)** Genomic DNAs from the designated strains grown in the specified media were tested for C-circle levels. The blot is displayed at the top and the quantitation of data from two independent measurements shown at the bottom. **(E)** Genomic DNAs from the designated strains grown in the specified media were assessed for the levels of G- and C-strand ssDNA using in-gel hybridization analysis (left panels). The gels were also denatured and re-probed to characterize the distribution of all telomere-containing fragments (right panels). **(F)** The levels of ssDNA as well as total DNA (from native and denatured gels, respectively) for the *blm^crg1* clones grown in YPA (for 4, 24, 48, and 72 hours) were quantified and plotted separately for the C-strand and G-strand signals. For each series, the quantified values were normalized to the timepoint with maximal values. Data (mean ± SEM) were from two independent experiments.

of Trf2 inhibition. That is, the phenotypes of Trf2 deficiency may be due to the combined actions of multiple DNA repair factors rather than Blm alone.

## Blm overexpression-induced telomere defects are counter-balanced by Trf2 overexpression

The ability of Blm overexpression to trigger telomere defects suggests an *in vivo* assay for assessing Blm inhibition by Trf2. In particular, we reasoned that if Trf2 is indeed a direct Blm inhibitor, then overexpressing Trf2 may counteract the effects of Blm overexpression. To test this possibility, we engineered a series of single and double mutants harboring the *blm^nar1^* and *trf2^crg1^* allele, which can be independently overexpressed in nitrate-containing and arabinose-containing media, respectively (Fig 5A). The strains were then grown in YPA to overexpress Trf2 only, and in MMA (minimal media with nitrate and arabinose) to overexpress both Blm and Trf2. RT-qPCR analysis confirmed that the strains harboring the *blm^nar1^* allele overexpressed blm RNA in MMA (nitrate plus) and repressed *blm* RNA levels in YPA (nitrate minus) (Fig 5B). We first assessed the ability of Trf2 overexpression to antagonize Blm overexpression using C-circle analysis. As shown in Fig 5C, Blm overexpression alone (*blm^nar1^* grown in MMA) induced high levels of C-circles, whereas concurrent overexpression of Trf2

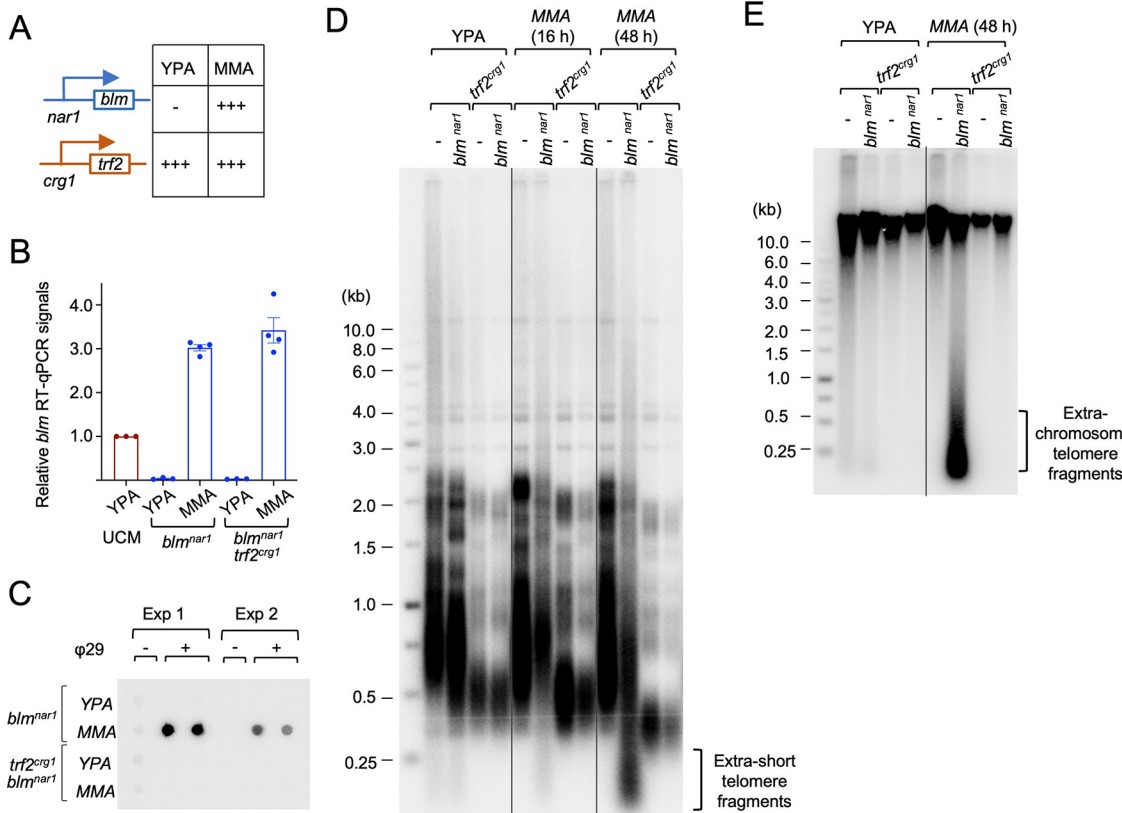

**Fig 5. The consequences of overexpressing Blm, Trf2 and both. (A)** The transcription activities of the *crg1* and *nar1* promoter in different media are schematically illustrated. **(B)** RT-qPCR analysis of *blm* expression in strains carrying the *blm^nar1^* allele grown in different media. Data (mean ± SEM) are from three or more independent measurements. **(C)** Genomic DNAs from the designated strains grown in YPA and MMA were subjected to C-circle analysis. The assay results from two independent experiments are shown. **(D)** Genomic DNAs from the designated strains grown in YPA and MMA were digested with *Pst*I and subjected to telomere Southern analysis. The extra-short telomeres triggered by Blm overexpression are marked by a bracket to the right of the blot. **(E)** Genomic DNAs from the designated strains grown in YPA and MMA were analyzed by telomere Southern without prior *Pst*I digestion. The extra-chromosomal telomeres triggered by Blm overexpression are marked by a bracket to the right of the blot.

($blm^{nar1}$ $trf2^{crg1}$ grown in MMA) suppressed C-circle formation. We also examined the ability of Trf2 overexpression to counteract telomere length alterations triggered by Blm overexpression. Notably, the telomere length heterogeneity observed in $blm^{nar1}$ grown in nitrate-containing media is milder than that of $blm^{crg1}$ grown in arabinose-containing media, suggesting that $nar1$ may overexpress Blm to a lesser degree. Nevertheless, a substantial increase in extra-short telomere fragments can be detected in $blm^{nar1}$ grown in MMA after 48 h (Fig 5D, marked by a bracket). Consistent with suppression of Blm by Trf2, these extra-short telomere fragments were largely eliminated in the $blm^{nar1}$ $trf2^{crg1}$ mutant grown in MMA. Analysis of telomere signals using undigested DNA samples confirmed that the extra-short telomeres are extra-chromosomal fragments (Fig 5E). Together, these results further support inhibition of Blm activity by Trf2 *in vivo*. In addition, given that Blm and Trf2 were the only transcriptionally manipulated factors in the mutants, the inhibitory effect of Trf2 is likely to be direct.

## Opposing regulation of Blm by the two duplex telomere binding proteins in *U. maydis*

As noted before, in addition to Trf2, *U. maydis* harbors a fungi-specific duplex telomere repeat binding factor named Tay1 [26,27]. Previous studies suggest that instead of telomere protection, the main function of Tay1 is to stimulate Blm activity at telomeres in order to promote telomere replication and recombination [26]. Thus, *U. maydis* appears to achieve optimal regulation of Blm by utilizing two telomere proteins with opposing effects on this helicase. One potential implication of this is that loss of Tay1 function might dampen Blm activity and partially suppress defects engendered by Trf2 deficiency. Alternatively, Tay1 might have a secondary function in telomere protection that is only discernible in the absence of Trf2. To further investigate the interplay between Tay1 and Trf2 in telomere regulation, we compared the phenotypes of the $tay1\Delta$, $trf2^{crg1}$, and $tay1\Delta$ $trf2^{crg1}$ mutants (Fig 6). Because we had previously generated $tay1\Delta$ using the FB1 parental strain, we proceeded to construct and analyze both $trf2^{crg1}$ and $tay1\Delta$ $trf2^{crg1}$ in the same strain background.

Growth analysis revealed similar growth defects for $trf2^{crg1}$ and $tay1\Delta$ $trf2^{crg1}$ on YPD, suggesting that Tay1 does not alter the toxicity induced by Trf2 deficiency (Fig 6A). However, TRF Southern analysis uncovered several interesting differences between the single and double mutants (Fig 6B). First, when grown in YPA, the $tay1\Delta$ $trf2^{crg1}$ double mutant manifested shorter telomeres than either $tay1\Delta$ and $trf2^{crg1}$ single mutants, indicating that the telomere shortening induced by these mutants are not mediated by identical mechanisms. Second, when grown in YPD, the $tay1\Delta$ $trf2^{crg1}$ double mutant manifested less severe telomere aberrations than the $trf2^{crg1}$ single mutant, consistent with mild suppression of telomere deprotection by $tay1\Delta$–presumably through the reduction of Blm helicase activity (Fig 6B). We reasoned that this suppression could be more prominent if Trf2 is only partially lost such that Blm is not overactive. Indeed, when spotted on plates containing 1.5% arabinose and 1% glucose (to enable partial expression of the $trf2^{crg1}$ allele), $tay1\Delta$ $trf2^{crg1}$ manifested improved growth and substantially milder telomere aberrations in comparison to $trf2^{crg1}$ (S6 Fig). Together, these observations highlight the opposing effects of Tay1 and Trf2 on Blm function at telomeres and suggest that Tay1 does not serve any function in telomere protection even in the absence of Trf2.

## Aberrant transcription of G- and C-strand telomere repeats in Blm-overexpressing and Trf2-deficient cells

Our recent analysis of Pot1-deficient cells indicates that upon telomere deprotection, both the G- and C-strand of telomere repeats are transcribed at abnormally high levels, suggesting that

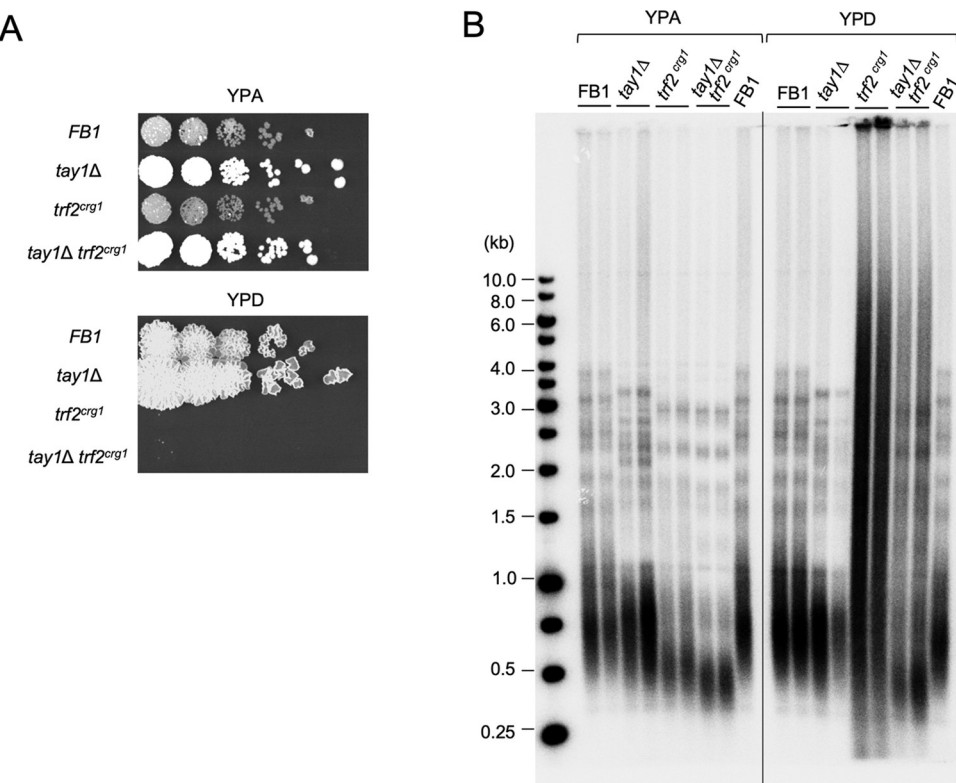

**Fig 6. The relationships between Tay1 and Trf2 in telomere protection. (A)** Serial dilutions of the indicated *U. maydis* strains were spotted onto YPA or YPD plates and incubated at 30°C. The plates were imaged following 3 days of growth. **(B)** DNAs from the designated strains grown in the specified media were subjected to TRF Southern analysis.

deprotected telomeres are substrates for the transcription apparatus [33]. There is also growing evidence that DNAs flanking DSBs are actively transcribed, pointing towards another resemblance between DSBs and deprotected telomeres [34–36]. To determine if aberrant transcription is associated with deprotected telomeres in general, we examined the levels of G- and C-strand telomere repeat RNAs in Blm-overexpressing mutants (*blm*crg1 grown in YPA) using both dot blot and RT-qPCR analyses (Fig 7A and 7B). Consistent with elevated transcription at de-protected telomeres, we found high levels of both G- and C-strand RNAs in *blm*crg1 strains grown in YPA but not YPD (Fig 7A and 7B). Interestingly, quantifications of the results point to higher levels of C-strand RNAs in comparison to G-strand RNAs (by ~ 10 and 50-fold as assessed by dot blot and RT-qPCR, respectively). We also used endpoint RT-PCR to assess the structures of these RNAs, which confirmed the existence of G- and C-strand RNAs that span both subtelomeric and telomeric regions (Fig 7C). For example, using forward and reverse primers that anneal to the subtelomeric and telomeric regions, respectively, we successfully amplified C-strand derived products that could only come from RNAs containing both regions (Fig 7C, middle panel). As another test of the hypothesis that telomere deprotection triggers aberrant transcription, we examined the levels of G- and C-strand RNAs in Trf2-deficient cells (S7 Fig). Indeed, both *trf2*crg1 and *tay1Δ trf2*crg1 manifested high levels of telomere repeat RNAs when grown in YPD. As in the case of Blm-overexpressing cells, the C-strand RNAs appear to be present at significantly higher levels as judged by RT-qPCR analysis (S7B Fig) Together, these findings indicate that aberrant transcription of both strands of the

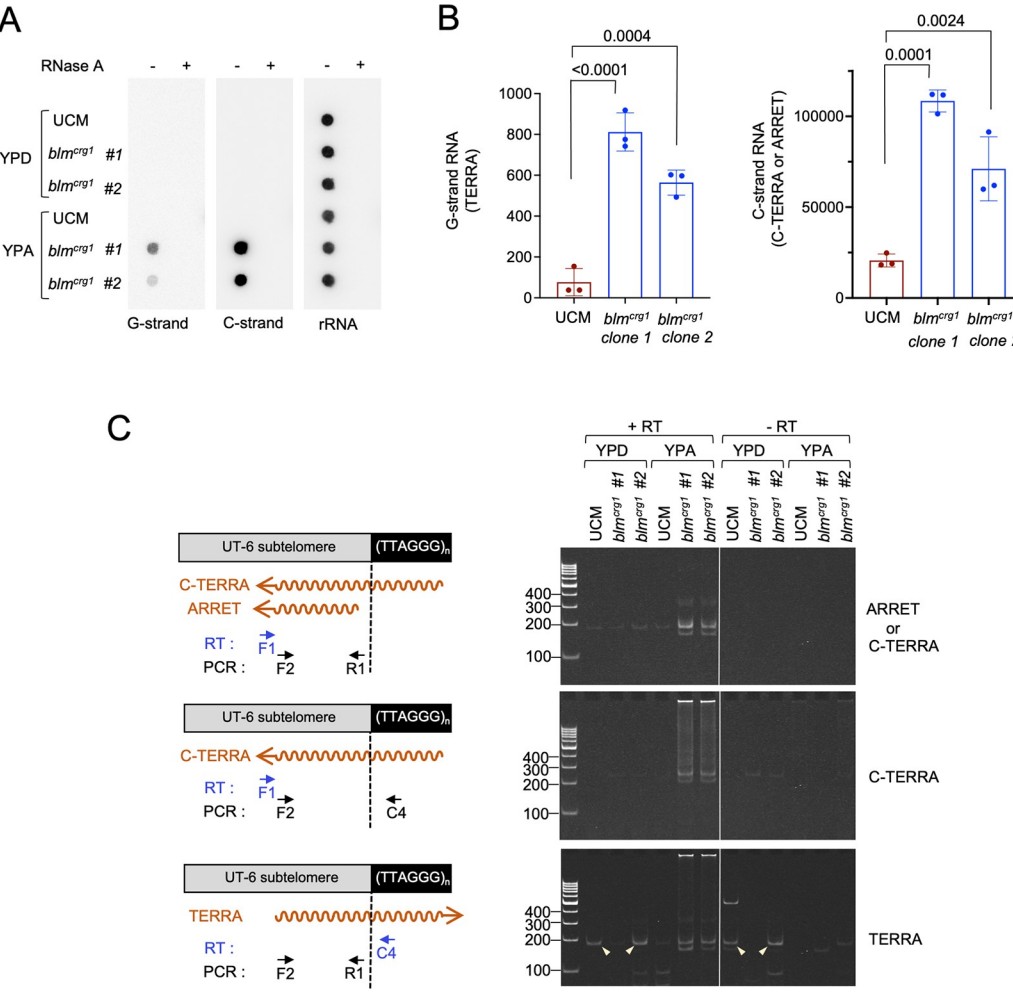

**Fig 7. Elevated telomere G- and C-strand RNA synthesis in Blm-overexpression mutants. (A)** RNAs were prepared from UCM and *blm^crg1* grown in YPA or YPD and spotted onto nylon filters with and without prior RNase A digestion. The filters were probed sequentially for G-strand RNA, C-strand RNA, and rRNAs. **(B)** RNAs were isolated from UCM and *blm^crg1* grown in YPA or YPD, and subjected to RT-qPCR analysis to quantify the levels of G- and C-strand RNAs. For the G-strand RNA, the primers were designed to amplify TERRA (RNAs spanning subtelomeres and telomeres) from the UT6-bearing telomeres. For C-strand RNA, the primers can in principle amplify both C-TERRA (RNAs spanning subtelomeres and telomeres) and ARRET (RNAs with subtelomere sequences only). The data are from three independent measurements and the values plotted denote cDNA copy numbers from 1 μl of RT reactions. The samples were compared using ANOVA with Dunnett's multiple comparisons test and the p values displayed. **(C)** RNAs were isolated from the indicated strains grown in the specified media, and subjected to endpoint RT-PCR analysis to characterize the structures of the G- and C-strand RNAs. Note that the TERRA assays for the YPD samples produced some RT-independent PCR products (marked by arrowheads), suggesting that these samples are contaminated with DNA. However, the YPA samples clearly generated RT-dependent products that were specific for the *blm^crg1* strains and absent from the UCM control.

telomere repeat region is a shared feature of de-protected telomeres triggered by aberrations in multiple telomere proteins including Pot1, Trf2 and Blm.

## Discussion

Studies of mammalian TRF2, one of two major duplex telomere repeat binding factors, have highlighted its critical roles in suppressing ATM signaling and cNHEJ at chromosome ends. In this study, we interrogated the functions of *trf2* in the basidiomycete fungus *U. maydis* and

uncovered a major function for this protein in suppressing Blm-driven telomere recombination at chromosome ends. Notably, several previous investigations of mammalian TRF2 also hint at a physical and functional connection between TRF2 and BLM [17,18]. Below we discuss the mechanistic and evolutionary implications of our findings.

## The regulation of Blm by Trf2; functional connections between Trf2 and Ku70/Ku80 complex

Our data argue that a primary function of *U. maydis* Trf2 is to suppress Blm activity at telomeres; all of the major telomere aberrations observed in the *trf2*-deficient setting are eliminated by deletion of *blm*. This is quite unexpected and provides another striking illustration of the concept that telomeres have evolved to antagonize aberrant repair. Notably, the phenotypes of *trf2*-deficient cells are quite similar to those triggered by loss of *U. maydis* Ku70 and Ku80, including telomere length heterogeneity, accumulation of ssDNA, and high levels ECTRs [30,37]. Moreover, these phenotypes of the *trf2* and *ku70/ku80* mutants are both strongly suppressed by *blm* or *mre11* deletion ([30] and this study). These findings suggest a physical connection or functional overlap between Trf2 and Ku70/Ku80. Indeed, previous analyses of human telomeres have uncovered an interaction between these proteins and implicated this interaction in the regulation of NHEJ at telomeres [14,38]. Therefore, the existence of such a direct interaction in *U. maydis* warrants further investigation. However, it is also worth noting that the *trf2* and *ku70/ku80* mutants are not phenotypically identical. In particular, whereas *ku70/ku80* loss results in high levels C-strand ssDNA, Trf2 deficiency leads to the accumulation of G-strand. Therefore, either Trf2 or Ku70/Ku80 must be capable of independently acting on some DNA repair factors to alter resection or recombination. The distinct ssDNA accumulation thus provides a phenotypic assay to dissect the mechanistic distinctions between Trf2 and the Ku complex at telomeres.

The viability of the *trf2*^*crg1* *blmΔ* and *trf2*^*crg1* *mre11Δ* mutants has striking implications and raises some interesting issues. One implication relates to the DNA damage signaling pathway (s) triggered by Trf2 loss: given the viability of *trf2*^*crg1* *blmΔ* and *trf2*^*crg1* *mre11Δ*, this signaling pathway is either inactive or unable to arrest growth in the absence of Blm or Mre11. Therefore, DNA damage signaling and DNA repair appear to work in a mutually reinforcing manner in the context of telomere protein deficiency to trigger telomere de-protection and growth arrest. Unfortunately, while a number of DDR kinases have been identified in *U. maydis*, relatively little is known about their activities at *U. maydis* telomeres. In this regard, it is worth noting that both *atr1Δ* and *chk1Δ* can rescue the lethality of Ku70-deficiency, but only partially suppress the telomere defects of this mutant [30,37]. This result suggests that Blm may act upstream of *atr1* and *chk1* to instigate telomere deprotection and checkpoint activation.

An unresolved question regarding the *trf2*^*crg1* *blmΔ* and *trf2*^*crg1* *mre11Δ* mutants is the nature of the telomere nucleoprotein structure in these mutants when Trf2 is depleted. Telomeres are substantially shorter (due to inefficient telomere replication in the absence of Blm or Mre11), but it is unclear if Tay1, the other duplex telomere binding protein, can fully occupy the telomere repeat tracts. Could nucleosomes bind to a significant portion of the telomeres, and what are the impacts of such an altered nucleoprotein structure on telomere protection and maintenance? Also unclear is whether changes at duplex telomere repeats can affect the nucleoprotein structure (e.g., the binding of Pot1 and Tpp1) at the single-stranded telomere termini. Interestingly, even though *trf2*^*crg1* *blmΔ* and *blmΔ* exhibit similarly short telomeres in YPD, the former manifested slower growth, suggesting that the altered telomeres devoid of Trf2 may impinge on cell cycle regulation. Further analysis of the structure and nucleoprotein composition of telomeres in the *trf2*^*crg1* *blmΔ* and *trf2*^*crg1* *mre11Δ* mutants should be informative.

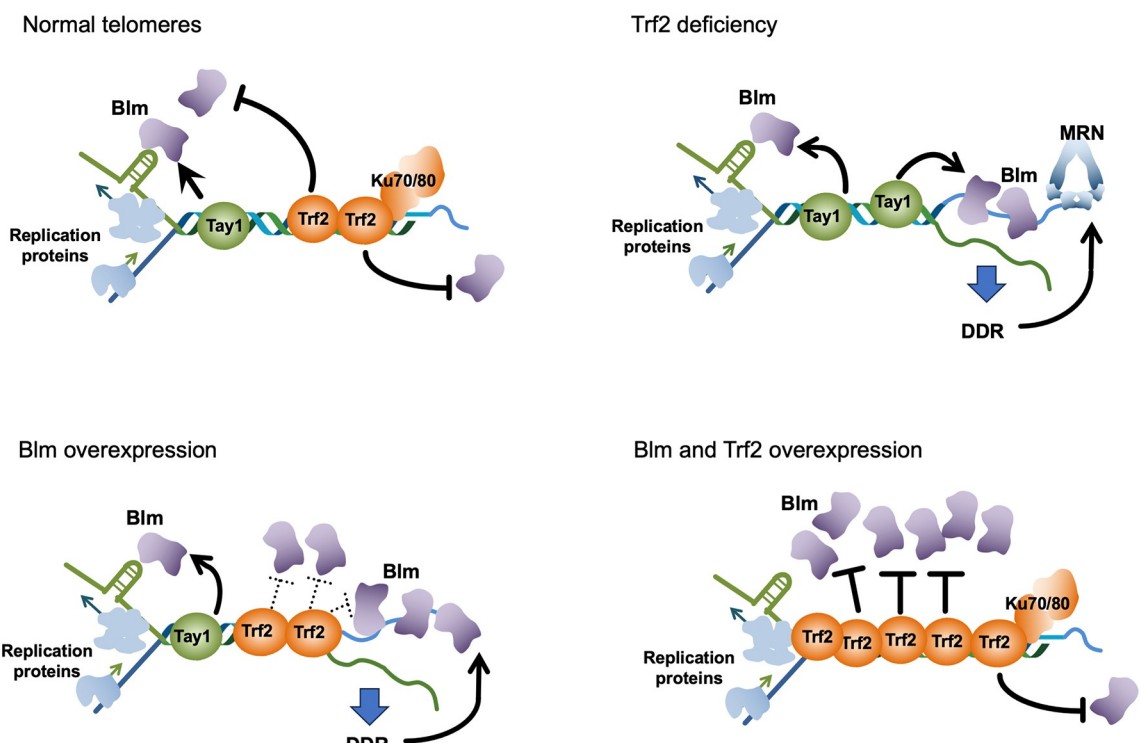

**Fig 8. A model for the interrelationships between telomere proteins and DNA repair factors analyzed in this study.** At normal telomeres (upper left), Blm is stimulated by Tay1 to promote telomere replication but suppressed by Trf2 and Ku70/80 to prevent telomere de-protection. When Trf2 is depleted (upper right), Blm becomes unopposed and mediates high levels of resection and recombination in conjunction with the MRN complex. Similarly, overexpression of Blm (lower left) can override Trf2 mediated telomere protection, leading to DDR and aberrant telomere repair. However, when Blm and Trf2 are both overexpressed, Trf2 can restore normal balance by suppressing the stimulatory activity of Tay1 and inhibiting Blm directly from acting at telomeres.

## Opposing regulation of Blm by duplex telomere repeat binding proteins

A key observation of the current study is opposing regulation of Blm by two duplex telomere repeat binding proteins (i.e., Tay1 and Trf2), which underscores the complex roles of Blm at telomeres (Fig 8). The rationale for the antagonism between Tay1 and Trf2 is evidently to fine-tune Blm activity such that this helicase is able to promote telomere replication without triggering abnormal repair. The need to balance the activating and repressing function of Tay1 and Trf2 vis-à-vis Blm is further illustrated by the dosage effects of both Trf2 and Blm at telomeres. Excessive suppression of Blm via Trf2-overexpression leads to telomere shortening. In contrast, over expression of Blm can override Trf2-mediated telomere protection, unless Trf2 is concurrently overexpressed. Therefore, the optimal calibration of Blm function is also predicated on expressing appropriate levels of Blm and Trf2 proteins. In short, an elaborate system has evolved in *U. maydis* to optimize Blm function, based on proper levels of Tay1, Trf2 and Blm, as well as opposing effects of Tay1 and Trf2 on Blm activity.

## Physical and function interactions between mammalian TRF2 and BLM; potential similarities between U. maydis and mammals

Even though mammalian TRF2 has not been reported to suppress BLM activity, this possibility is hinted by several previous findings. First, overexpression of TRF2 in primary cells increased the rate of telomere shortening, which could be due to reduced efficiency of telomere

replication as a result of BLM inhibition [39]. Second, overexpression of TRF2 in telomerase-positive cancer cells caused replication fork stalling at telomeres and induced the formation of telomeric ultrafine bridges, which may also be explained by compromised BLM function [40]. Indeed, anaphase ultrafine bridges are characteristic of BLM-deficient cells [41]. Interestingly, in contrast to *Um*Trf2, mammalian TRF2 was previously reported to stimulate BLM helicase activity *in vitro* [17,18]. This discrepancy could be due to differences in reaction components or conditions between the earlier and current studies, e.g., in the lengths of telomere repeats in substrate DNAs. Recently, human TRF2 was found to cooperate with RAP1 to regulate multiple BLM-mediated reactions *in vitro* [42]. Additional studies in mammalian cells, especially regarding the *in vivo* effect of TRF2 expression on BLM activity, may be informative.

As noted before, the predominant consequence of TRF2 dysfunction at mammalian telomeres is NHEJ, whereas that of Trf2 deficiency at *U. maydis* telomeres is Blm-mediated resection and recombination. This difference, however, does not necessarily argue against a BLM inhibitory function for mammalian TRF2. One possibility is that TRF2 may work redundantly with another factor to restrain BLM activity. Alternatively, NHEJ may be less active at *U. maydis* telomeres, leading to preferential channeling of Trf2-deficient telomeres through the Blm-dependent recombination pathway.

## Elevated transcription at de-protected telomeres

We found that similar to the *pot1* mutant, in both Blm-overexpressing and Trf2-deficient cells, the telomere de-protection is accompanied by elevated levels of both G- and C-strand telomere repeat RNAs [33]. These findings suggest that increased transcription of both telomere strands is a general consequence of telomere de-protection. This in turn raises a key question regarding the nature and consequences of these transcripts, namely, do they represent elevated levels of RNAs that are normally expressed at low levels, or do they represent fundamentally abnormal RNAs that are produced only in pathologic settings? In this regard, it is worth noting that most studies of cells with normal telomere structures have revealed G-strand RNAs that initiate from subtelomeric promoters and elongate into telomere repeats (i.e., TERRA) [43,44]. C-strand RNAs, in contrast, have not been reported or characterized in normal cells except in the case of *S. pombe*; whether these C-strand RNAs perform important physiologic functions remains unclear [45,46]. Thus, it seems likely that at least a portion of the RNAs in the mutants represent fundamentally different transcripts that are only synthesized from deprotected telomeres.

While more studies will be required to understand the nature of the telomere transcripts in the mutants, we note that the synthesis of such aberrant RNAs exhibits similarities to two previously described RNA species. First, there is growing evidence that regions flanking double strand breaks (DSBs) are transcribed, and that these RNAs form RNA-DNA hybrids that can modulate the repair of DSBs [34–36]. Since dysfunctional telomeres induce similar cellular response as that by double strand breaks, the telomere transcripts may be synthesized through the same pathway(s) as the DSB-associated RNAs. In support of this idea, the DSB-associated RNAs are thought to correspond primarily to the strand that bears the 5' ends, i.e., equivalent to the C-strand [35]. Therefore, the DSB paradigm provides a plausible explanation for why the *U. maydis* telomere protein mutants harbor higher levels of C-strand RNAs. Alternatively, the abnormal telomere transcripts in the mutants could be a consequence of perturbing telomere chromatin structure, much as global depletion/perturbation of nucleosomes has been shown to trigger promiscuous transcription [47,48]. In this paradigm, the synthesis of aberrant telomere RNAs is unrelated to the presence of the DNA ends and is simply governed by the accessibility of DNA to the transcription apparatus. Regardless of the mechanisms of telomere

RNA synthesis, once generated, these RNAs have the capacity to form hybrids and R-loops with telomere repeats, thereby altering DNA damage response and DNA repair. Hence a full understanding of telomere dysfunction requires insights not only on the synthesis of aberrant RNAs, but also their impacts on telomere structure and function. The conditional mutants that we have developed in *U. maydis* provides a useful system to interrogate these issues.

## Materials and methods

### *Ustilago maydis* strain construction and growth conditions

Standard protocols were employed for the genetic manipulation of *U. maydis* [49–51]. All *U. maydis* strains used in this study were haploid and were derived from the UCM350 or FB1 background [50,52] and are listed in S1 Table. The growth media used included YPA (2% peptone, 1% yeast extract, 1.5% arabinose), YPD (2% peptone, 1% yeast extract, 2% glucose), and MMA [0.3% $KNO_3$, 0.062%(v/v) salt solution, 1.5% arabinose]. Media with arabinose and glucose were used for the induction and repression of the *crg1* promoter, respectively. Media with nitrate (MMA) or without nitrate (YPA and YPD) were used for the induction and repression of the *nar1* promoter.

The *blm^crg1* strain was constructed by integrating a linearized pRU12-*blm* plasmid containing fragments that correspond to (1) 700 bp of the *blm* promoter and (2) the first 700 bp of the *blm* ORF into the *blm* genomic locus in UCM350. Briefly, the two *blm* fragments were generated by PCR using appropriate oligos with flanking restriction enzyme sites (S2 Table), and then cloned together in between the *Nde*I and *Eco*RI sites of pRU12 [28]. Correct recombination of *Xba*I treated pRU12-*blm* into the genomic locus results in the insertion of the *cbx^R* marker and the *crg1* promoter precisely adjacent to and upstream of the *blm* ATG initiation codon. Following transformation, the correct mutant strains (*blm^crg1*) were selected on YPA supplemented with carboxin (4 μg/ml) and confirmed by PCR. The *blm^nar1* strain was similarly constructed using a linearized (digested by *Nsi*I) pRU2-Hyg-*blm* plasmid carrying identical *blm* promoter and coding regions. The transformants were selected on YPA supplemented with hygromycin (150 μg/ml) and confirmed by PCR. The *trf2^crg1 blm^nar1* and *pot1^crg1 blm^nar1* mutants were made by transforming *trf2^crg1* and *pot1^crg1* with the linearized pRU2-Hyg-*blm* plasmid and screening for correct transformants. The *blmΔ trf2^crg1*, *dna2Δ trf2^crg1*, *exo1Δ trf2^crg1*, and *mre11Δ trf2^crg1* mutants were constructed by transforming XbaI digested pRU12-Trf2 into the respective deletion mutants, and screening for correct transformant by PCR genotyping. The *rad51Δ trf2^crg1* mutant was made by transforming *trf2^crg1* with a *rad51* disruption cassette (pUC-M2T1-rad51 digested with *Ssp*I).

For growth analysis of the mutants, liquid cultures of *U. maydis* strains were first diluted to $OD_{600}$ of 1.0 (corresponding to ~ 0.5 x $10^6$ cells/ml), and subjected to serial 6-fold dilutions. Four microliters of the diluted cultures were then spotted onto YPA or YPD plates and grown at 30˚ C for 2 to 3 days.

### RT-qPCR analysis of *blm* and *trf2* RNA levels

For RT-qPCR analysis, RNAs were extracted and purified from log-phase *U. maydis* cultures using acid phenol extraction [53]. The samples were treated with TURBO DNase (Thermo Fisher) and then reverse transcribed in 20 μl reactions that contained 1x SSIV Buffer, 12.5 μM primer (S2 Table), 1 mM dNTP, 5 mM DTT, 20 U NxGen RNase Inhibitor (Lucigen), and 40 U Superscript IV RT (Thermo Fisher). The reaction mixtures were incubated at 55˚ C for 40 min and then heated at 70˚ C for 15 min to inactivate the reverse transcriptase. qPCR was carried out in Chai Open qPCR Dual Channel thermocycler using Chai Green Master mix with 8 μM primers (S2 Table) and 1 μl of cDNA. A series of copy number standards (*blm* and *trf2*)

were prepared by amplifying regions of the genes or subtelomeres that span the qPCR primers. Cycling parameters were as follows: 45 cycles of 95˚C for 30 sec, 53˚C for 20 sec, 72˚C for 40 sec.

## Telomere length and structural analyses

Genomic DNAs were isolated from cells grown in liquid cultures. Unless specifically indicated in the figures, the cells were cultured in non-permissive media for 16 to 24 hours before harvesting and DNA isolation. Southern analysis of telomere restriction fragments (TRF Southern) was performed using DNA treated with *Pst*I as described previously [25,29]. The blots were hybridized to an oligonucleotide containing eight copies of the TTAGGG repeats. For quantitation, we used the formula mean TRF length = $\Sigma$ (OD$_i$ × L$_i$) / $\Sigma$ (OD$_i$) because the majority of TRF length variation stems from differences in subtelomeric lengths [54,55]. The in-gel hybridization assays for detecting ssDNA on the G- and C-strand have been described [37]. STELA for telomeres bearing UT4 subtelomeric elements was performed essentially as reported before [31]. C-circle and G-circles assays were performed as before except that NxGen phi29 DNA Polymerase (Lucigen Corp.) were used for rolling circle amplifications [25].

## Analysis of telomere repeat-containing RNAs

RNAs were purified from log-phase *U. maydis* cultures using acid phenol extraction [53]. For dot blot analysis, 0.2 μg of RNAs (untreated or pre-treated with various nucleases) in 10 μl were diluted with 30 μl of RNA incubation/denaturation solution (65.7% formamide, 7.7% formaldehyde, 1.3x MOPS buffer), heated at 65˚C for 5 min, and then cooled on ices. After adding 40 μl of 20x SSC, the entire samples were applied to Hybond-N using a Bio-Dot Microfiltration Apparatus (Bio-Rad Laboratories, Inc.). After UV crosslinking, the blot was probed sequentially for G-strand and C-strand RNAs using labeled C8 and G8 oligonucleotides, respectively. As a further loading control, the blot was stripped and re-probed with a labeled PCR fragment corresponding to a region of *U. maydis* 26S rRNA. For end-point RT-PCR, the RNA samples from acid phenol extraction were first treated with Turbo DNase (Thermo Fisher Inc.). Fifty ng of RNAs were subjected to reverse transcription using Superscript III RT and appropriate primers (UT6-TERRA-F1 for C-strand RNAs and C4 for G-strand RNAs) at 55˚C for 40 min, followed by heating at 70˚C for 15 min to inactivate R. The cDNA products were then subjected to PCR (33–37 cycles of 95˚C for 15 sec, 66˚C for 15 sec, and 72˚C for 30 sec) with Q5 polymerase (NEB Inc.) and appropriate primers (S2 Table). For RT-qPCR, the same cDNA products were analyzed by qPCR using UT6 primers (UT6-TERRA-F2V2 and UT6-TERRA-R2V2) and the following parameters: 45 cycles of 95˚C for 30 sec, 56˚C for 20 sec, 72˚C for 40 sec.

## Pull down and helicase assays for analyzing Trf2-Blm interactions

Blm-FG was first expressed as His$_6$-SUMO-Blm-FG fusion (Blm with an N-terminal His$_6$-SUMO tag and a C-terminal FLAG tag) in the BL21 codon plus strain and purified by a combination of (i) Ni-NTA chromatography, (ii) ULP1 cleavage, and (iii) anti-FLAG affinity chromatography as previously described [25]. Following binding to the anti-FLAG resin and washing with FLAG(150) buffer (50 mM Tris-HCl, pH 7.5, 150 mM NaCl, 10% glycerol, 0.1% NP-40, 2.5 mM MgCl$_2$, 1 mM DTT), the beads were used directly in pull down assays. The prey for these assays, the His$_6$-SUMO-Trf2$^{\Delta N}$ fusion (abbreviated as His-Trf2$^{\Delta N}$), was also expressed BL21 and purified via Ni-NTA chromatography. The partially purified His-Trf2$^{\Delta N}$ was then mixed with anti-FLAG resin carrying immobilized Blm-FG in FLAG(65) buffer

(same as FLAG(150) except that NaCl is at 150 mM). Following incubation with mixing on a rotator at 4˚C for 1 h, the beads were washed 4 times with 0.5 ml of the FLAG(100) buffer (same as FLAG(150) except that NaCl is at 100 mM), and then the bound proteins eluted with 60 µl FLAG(150) containing 0.2 mg/ml FLAG$_3$ peptide. The eluates were analyzed by Western Blot using anti-His and anti-FLAG antibodies.

The helicase assays were performed as described using 25 nM Blm-FG and varying concentrations of Trf2$^{\Delta N}$ [26]. After 20 min of incubation at 35˚C, the reactions were terminated by the addition of 4 µl stop solution (30% glycerol, 50 mM EDTA, 0.9% SDS, 0.1% BPB and 0.1% xylene cyanol) and 20 ng of unlabeled bottom strand oligo (as trap to prevent reannealing of unwound ssDNA) or directly loaded onto 10% polyacrylamide gels (acrylamide: bis-acrylamide = 29: 1) containing 1x TBE. The substrates and products were separated via electrophoresis, and the results analyzed using a Phosphor scanner. The ratio of labeled DNA in single-stranded form (bottom species in the gel) to total DNA (all the labeled species) at the end of the reaction is taken as the fraction unwound by Blm helicase.

## Statistical analysis

Statistical analyses were performed using GraphPad Prism 10.0. Comparison of multiple samples were carried out using ANOVA with Dunnett's multiple comparisons test.

## Supporting information

**S1 Fig. Growth defects of the *trf2$^{crg1}$ blmΔ* and *trf2$^{crg1}$ mre11Δ* mutants grown in YPD. A.** Serial dilutions of the indicated strains were spotted onto YPD medium. The growth of the strains was imaged after 2 days and 3 days. **B.** The indicated strains were inoculated into fresh YPD at an OD$_{600}$ of 0.01 and grown at 30 degree for 17 hours. The fold increases in OD$_{600}$ for two independent cultures were calculated and plotted.
(TIF)

**S2 Fig. Phenotypic analysis of Trf2-deficient cells derived from the FB1 strain background. A.** Serial dilutions of the indicated strains were spotted onto YPA and YPD media and incubated at 30˚C. Following 2 days of growth, the plates were imaged. **B.** Genomic DNAs from the indicated strains were subjected to TRF Southern analysis.
(TIF)

**S3 Fig. A progressive reduction in Trf2 expression level correlates with a gradual increase in telomere lengths. A.** UCM and *trf2$^{crg1}$* were grown in media containing 1.5% arabinose and varying concentrations of glucose (to reduce the activity of the *crg1* promoter). RNAs were isolated from these cultures and subjected to RT-qPCR analysis to determine the relative *trf2* RNA levels. Data (mean ± SD) were derived from three or more independent experiments. **B.** Genomic DNAs were isolated from UCM and *trf2$^{crg1}$* grown in media with the specified arabinose and glucose concentrations and subjected to TRF Southern analysis.
(TIF)

**S4 Fig. Epistasis analysis of telomere shortening mediated by Trf2-overexpression and deletions of *exo1*, *blm*, *mre11*, and *dna2*. A and B.** Genomic DNAs were isolated from the indicated strains grown in YPA, digested with *Pst*I, and subjected to TRF Southern analysis.
(TIF)

**S5 Fig. Blm overexpression triggers ECTRs in a reversible manner. A.** DNAs were isolated from *blm$^{crg1}$* strains grown in the specified culture media and subjected to Southern analysis for telomere DNA with and without prior *Pst*I digestion. **B.** DNAs were isolated from UCM

and *blm*[crg1] strains grown in the designated culture media and subjected to Southern analysis. To test the reversibility of telomere defects induced by Blm overexpression, the *blm*[crg1] clones were first grown on a YPA plate and then re-streaked once on a YPD plate prior to liquid culture growth and genomic DNA isolation (designated by YPA -> YPD).
(TIF)

**S6 Fig. Partial suppression of Trf2-deficiency phenotypes by *tay1Δ*. A.** Serial dilutions of the designated strains were spotted onto semi-solid media containing the specified combinations of arabinose and glucose, and grown for 2 or 3 days at 30 °C. **B.** Genomic DNAs were isolated from the indicated strains grown in 1.5% arabinose and 1% glucose, and subjected to TRF Southern analysis.
(TIF)

**S7 Fig. Up-regulation of telomere G- and C-strand RNA synthesis in Trf2-deficient mutants. A.** RNAs were isolated from the indicated strains grown in YPD and spotted onto nylon membrane. After crosslinking, the membrane was probed sequentially for G-strand RNA, C-strand RNA, and rRNAs. **B.** RNAs were isolated from the indicated strains grown in YPD, and subjected to RT-qPCR analysis to quantify the levels of G- and C-strand RNAs. For the G-strand RNA, the primers were designed to amplify TERRA (RNAs spanning subtelomeres and telomeres) from the UT6-bearing telomeres. For C-strand RNA, the primers can in principle amplify both C-TERRA (RNAs spanning subtelomeres and telomeres) and ARRET (RNAs with subtelomere sequences only) from UT6 telomeres. The quantities plotted (mean ± SD) represent cDNA copy numbers from 1 μl of RT reactions from two independent experiments.
(TIF)

**S1 Table. *U. maydis* strains used in this study.**
(DOCX)

**S2 Table. Oligonucleotides used in this study.**
(DOCX)

**S1 Data. The numerical data used for all plots in the paper.**
(XLSX)

## Author Contributions

**Conceptualization:** Shahrez Syed, William K. Holloman, Neal F. Lue.

**Data curation:** Shahrez Syed, Sarah Aloe, Neal F. Lue.

**Formal analysis:** Shahrez Syed, Sarah Aloe, Neal F. Lue.

**Funding acquisition:** Neal F. Lue.

**Investigation:** Shahrez Syed, Sarah Aloe.

**Methodology:** Shahrez Syed, Sarah Aloe.

**Project administration:** Neal F. Lue.

**Resources:** Sarah Aloe, Jeanette H. Sutherland, William K. Holloman.

**Supervision:** Neal F. Lue.

**Writing – review & editing:** Shahrez Syed, William K. Holloman, Neal F. Lue.

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
