## [Decision Letter · Decision Letter 0]

17 Jun 2024

Dear Dr. Lue,

Thank you very much for submitting your Research Article entitled '*Ustilago maydis* Trf2 ensures genome stability by antagonizing Blm-mediated telomere recombination: fine-tuning DNA repair factor activity at telomeres through opposing regulations' to *PLOS Genetics*.

The manuscript was evaluated at the editorial level and by independent peer reviewers. Thank you kindly to you and all authors for your patience with the time needed for the thorough reviewing process. The reviewers appreciated the attention to an important problem, but raised some substantial concerns about the current manuscript. Based on the reviews, we will not be able to accept this version of the manuscript, but we would be willing to review a revised version. We cannot, of course, promise publication at that time.

Should you decide to revise the manuscript for further consideration here, your revisions should address the specific points made by each reviewer. Reviewer #2, in particular, lists several most-significant concerns that must be addressed. We will also require a detailed list of your responses to the review comments and a description of the changes you have made in the manuscript.

If you decide to revise the manuscript for further consideration at *PLOS Genetics*, please aim to resubmit within the next 60 days, unless it will take extra time to address the concerns of the reviewers, in which case we would appreciate an expected resubmission date by email to plosgenetics@plos.org.

To enhance the reproducibility of your results, we recommend that you deposit your laboratory protocols in protocols.io, where a protocol can be assigned its own identifier (DOI) such that it can be cited independently in the future. Additionally, *PLOS ONE *offers an option to publish peer-reviewed clinical study protocols. Read more information on sharing protocols at https://plos.org/protocols?utm_medium=editorial-email&utm_source=authorletters&utm_campaign=protocols

Please do not hesitate to contact us if you have any concerns or questions.

Yours sincerely,

David Zappulla, Ph.D.

Guest Editor

*PLOS Genetics*

Eva Stukenbrock

Section Editor

*PLOS Genetics*

Reviewer's Responses to Questions

**Comments to the Authors:**

*
(see also attached PDF file from Rev #1)
*

Reviewer #1: The article is best suited to an audience specialized in DNA replication and recombination at the end of chromosomes. This is a contribution to the knowledge of the mechanisms of maintaining telomere homeostasis.

Please also see the attachment.

In this article, the authors show the antagonistic control of UmTrf2 and Tay1 of Ustilago maydis on the activity of the Blm. The authors found that these telomere-binding proteins have different functions in telomere metabolism: UmTrf2 is essential, protects the chromosome end from NHEJ pathways of recombinational repair machinery, inhibits the Blm helicase activity, and has an epistatic relationship to it. Instead, Tay1 is not an essential protein in U. maydis; it plays a role in replication, promotes telomere recombination, and stimulates Blm activity. The deletion of UmTrf2 generates the accumulation of aberrant DNA forms due to an imbalance of the activities that generally occur at the chromosome end for its maintenance; such aberrant DNA production is alleviated by Blm and Mre11 deletion. Blm and Mre11 are needed for telomere replication and participate in the DNA resection of the chromosome end, indirectly controlling recombinational events. Antagonistic roles of UmTrf2 and Tay1 are demonstrated, as well as the finding of the previously reported TERRA and ARRET lncRNAs, but it was also found C-TERRA as a transcriptional product of dysfunctional chromosome ends in the designed experiments; notably, C-TERRA had not been identified previously except in the ascomycete Schizosaccharomyces pombe.

The work is splendid, sound, and well-founded, with original results showing features of the telomeric DNA protection mechanisms and how DDR by recombination repair systems operate in the basidiomycete U. maydis; this study also opens the possibility of investigating how telomere synthesis occurs or is controlled by lncRNAs under these circumstances.

The text is clear and focused; however, some minor details could be considered. In the numbered pages:

- Page 7. Last two rows: “However, subsequent analysis consistently revealed high levels of G-strand ssDNA, indicating that this is the reproducible phenotype of the mutant.” Could this mean “also revealed a high…”

- Page 9. Regarding exo1, “Additional analysis suggests that concerning telomere shortening, trf2O/E is also epistatic to exo1Δ, but not to dna2Δ (S4 Fig).” Please compare S4 Fig left vs. Fig.1B to avoid any matching problems.

- Page 11. For a broader audience, please consider indicating what it is MMA.

- Page 22. In subsection Pull down and helicase assays… (Ref.) needs attention, please cite, or if you agree, please tell the audience what the Blm FLAG was in the main text.

In the consecutive pages numbered by the Acrobat program

- Page 47. S5A Fig. To my eyes, the legends from clone1 and clone2 growth on YPA and YPD were interchanged, as those results do not match those of S5B Fig and Fig 4C. In the same image, please consider moving the PstI legend away from the MWM lane or indicating the presence of the enzyme above the lanes of digested DNA to avoid confusion.

- Page 48. As FB1 was used throughout the experiment, it would be good to homogenize the legends (WT or FB1(WT)) of the first row of each panel in S6A Fig.

- Page 36. In Fig. 7C, in the bottom right panels, please explain in the text what the red arrow is pointing at. Also, please explain why cDNA is amplified from transcripts from the UCM controls grown in YPD, as does in clone 2 (+ RT); please note that the same occurs even without reverse transcription (—RT) in the bottom right panel.

- Page 37. Please note the letters DDR are truncated in the image at the bottom left.

**Reviewer #2: **

In this manuscript, Zahid et al. examine the role of the telomere-binding protein Trf2 in repressing activity of the Blm helicase at telomeres in the fungal model organism Ustilago maydis. U. maydis telomeres are bound by Trf2, a protein orthologous to mammalian TRF2. Previous work has reported interactions between TRF2 and BLM helicase but the role of this interaction in telomere structure and function is unclear. Using genetic (knock-down and overexpression) studies and biochemical assays, the authors demonstrate that Trf2 negatively regulates Blm helicase and examine the opposing effects of telomere binding proteins Tay1 and Trf2 on Blm function at telomeres. The results contribute to the understanding of telomere biology and suggest that, at least in this organism, Blm repression is a critical function of Trf2. The genetic analysis would benefit from additional characterization of the conditional expression systems that the authors develop. Some additional information about statistical analyses is also required.

1. The conclusions of this manuscript are largely dependent on the conditional expression systems that are utilized. I recommend that the authors establish the extent of knock-down and overexpression up front (as a supplementary figure) prior to asking the reader to interpret the knock-down and overexpression phenotypes. Some data for trf2crg1 are included in Figure 3S where the extents of both expression changes are determined. Evidence of blm overexpression is shown in Figure 4A, but it is not clear how much expression is repressed on glucose. This is relevant because blm deletion causes telomere shortening (Figure 2A) but blmcrg1 on YPD has equivalent or even slightly longer telomeres than the WT strain. Why is this? The effect of the nar1 promoter on blm expression (used in Figure 5) should also be quantified and presented. In experiments where both blm and trf2 are under regulated control and the cells are grown on YPA, is blm expression repressed below normal levels?

2. Where error bars are present, the figure legend or methods should indicate what those error bars represent. When statistical analysis is shown, the test used should be indicated. For example, is a test that accounts for multiple comparisons utilized in Figure 2B and C?I

3. In cases where telomere lengths are analyzed in media that is non-permissive for colony formation, the length of time that cells have been grown in the indicated liquid medium should be stated.

4. The first paragraph of the Discussion section entitled “The regulation of Blm by Trf2; functional connections between Trf2 and Ku70/Ku80 complex” needs to be better referenced with regards to the U. maydis results for Ku70 and Ku80. Those results aren’t from this manuscript, are they? I don’t see them anywhere but there are no references.

5. Is it possible to include molecular weight markers on the native and denatured gels (Figure 1D, for example) to allow the reader to ascertain how the sizes correlate to those in the standard Southern blots (Figure 1B, for example).

6. Figure 2B is lacking a Y axis label.

7. In Figure 3A, “Pray” should be “Prey.” Is Blm-FG really cross-reacting, or is there a fraction of the protein that wasn’t cleaved during purification?

8. Error bars should be included on Figure 3C. How was quantification done? This information should be added to the figure legend.

9. Figure 4E, F: Is 0 hours in YPA the correct comparison, especially for the G strand signal? It looks like repression of blm expression in YPD strongly increases G strand signal so that the change upon overexpression is being exaggerated compared to the change relative to normal expression levels. These results should be more carefully described and discussed.

10. In Figure 5SA, the YPA and YPD labels are switched.

**Have all data underlying the figures and results presented in the manuscript been provided?**

Reviewer #1: Yes

Reviewer #2: Yes

PLOS authors have the option to publish the peer review history of their article (what does this mean?). If published, this will include your full peer review and any attached files.

Reviewer #1: No

Reviewer #2: No

---

## [Decision Letter · Decision Letter 1]

4 Nov 2024

PGENETICS-D-24-00128R1

Ustilago maydis Trf2 ensures genome stability by antagonizing Blmmediated telomere recombination: fine-tuning DNA repair factor activity at telomeres through opposing regulations

PLOS Genetics

Dear Dr. Lue,

Thank you for submitting your manuscript to PLOS Genetics. After careful consideration, we feel that it has merit but does not fully meet PLOS Genetics's publication criteria as it currently stands. Therefore, we invite you to submit a revised version of the manuscript that addresses the points raised during the review process.

Please submit your revised manuscript within 30 days. If you will need more time than this to complete your revisions, please reply to this message or contact the journal office at plosgenetics@plos.org. Please include the following items when submitting your revised manuscript:

*
A rebuttal letter that responds to the point raised by Reviewer 2. You should upload this letter as a separate file labeled 'Response to Reviewers'. This file does not need to include responses to formatting updates and technical items listed in the 'Journal Requirements' section below.

We look forward to receiving your revised manuscript.

Kind regards,

David C Zappulla, Ph.D.

Guest Editor

PLOS Genetics

Eva Stukenbrock

Section Editor

PLOS Genetics

Aimée Dudley

Editor-in-Chief

PLOS Genetics

Anne Goriely

Editor-in-Chief

PLOS Genetics

J**ournal Requirements:**

E**dditional Editor Comments (if provided):**

Dear Dr. Lue,

Thank you for your revised manuscript. The two reviewers have assessed the latest version. As you can see, Reviewer 1 states that it is now acceptable and Reviewer 2 is also mostly pleased with the revised manuscript, but noted just one remaining concern. Please address this last concern of Reviewer 2 about employing the appropriate statistical test(s) in Figures 2 and 7.

We look forward to receiving your revised manuscript.

Sincerely,

David

**Reviewers' comments:**

Reviewer's Responses to Questions

**Comments to the Authors:**

Reviewer #1: Dear Sir:

Authors of the article “Ustilago maydis Trf2 ensures genome stability by antagonizing Blm mediated telomere recombination: fine-tuning DNA repair factor activity at telomeres through opposing regulations” by Shahrez Syed, Sarah Aloe, Jeanette H. Sutherland, William K. Holloman, Neal F. Lue, have accomplished the suggestions that this reviewer made on the technical details and data presentation. The authors also fixed image troubles, clarified technical procedures, and improved the manuscript; hence, in the opinion of this reviewer, it satisfies the criteria for publication in the PLOS Genetics Journal.

The authors explore in an original approach the control of telomere maintenance and its sequence integrity by the interplay of Trf2 and helicase Blm with telomere repeats and possibly by a subtle interaction among those proteins to repress abnormal recombination events. In addition to the known involvement of Trf2 and Blm in telomere protection and replication/recombination, respectively, it is shown that severe imbalance in their appropriate expression rates causes aberrations in telomere sequence length and integrity, altered expression of ncRNAs from telomeric regions, loss of telomere homeostasis, and chromosome instability.

Reviewer #2: The authors have addressed most of my concerns and I appreciate the clarifications added to the manuscript. My one remaining concern relates to the multiple statistical comparisons that are done in Figures 2 and 7. It is not appropriate to conduct t-tests when the same sample is compared to multiple other samples in the same experiment. ANOVA with a post-hoc test would correct for multiple comparisons. For example, see https://jbcresources.asbmb.org/collecting-and-presenting-data.

**Have all data underlying the figures and results presented in the manuscript been provided?**

Reviewer #1: Yes

Reviewer #2: Yes

PLOS authors have the option to publish the peer review history of their article (what does this mean?). If published, this will include your full peer review and any attached files.

Reviewer #1: No

Reviewer #2: No

**Figure resubmission:**
---

## [Decision Letter · Decision Letter 2]

26 Nov 2024

Dear Dr Lue,

We are pleased to inform you that your manuscript entitled "Ustilago maydis Trf2 ensures genome stability by antagonizing Blm mediated telomere recombination: fine-tuning DNA repair factor activity at telomeres through opposing regulations" has been editorially accepted for publication in PLOS Genetics. Congratulations!

Yours sincerely,

David C Zappulla, Ph.D.

Guest Editor

PLOS Genetics

Eva Stukenbrock

Section Editor

PLOS Genetics

Aimée Dudley

Editor-in-Chief

PLOS Genetics

Anne Goriely

Editor-in-Chief

PLOS Genetics

Comments from the reviewers (if applicable):

Dear Dr. Lue,

Thank you for submitting your work to PLoS Genetics and for your patience with the review process.

I am pleased to inform you that Reviewer 2 was satisfied with your statistics for Figures 2 and 7. We are very pleased at this time to accept your manuscript for publication in PLoS Genetics.

Please note that Figure 8 is somehow after Figure 6 and before Figure 7 in the current manuscript as you prepare your documents for publication.

Sincerely,

David Zappulla

Reviewer's Responses to Questions

**Comments to the Authors:**

Reviewer #2: The authors have appropriately addressed my concern about the statistical analysis.

**Have all data underlying the figures and results presented in the manuscript been provided?**

Reviewer #2: Yes

PLOS authors have the option to publish the peer review history of their article (what does this mean?). If published, this will include your full peer review and any attached files.

Reviewer #2: No

**Data Deposition**

http://datadryad.org/submit?journalID=pgenetics&manu=PGENETICS-D-24-00128R2

**Press Queries**

---

## [Editor Report · Acceptance letter]

2 Dec 2024

PGENETICS-D-24-00128R2 

Ustilago maydis Trf2 ensures genome stability by antagonizing Blm mediated telomere recombination: fine-tuning DNA repair factor activity at telomeres through opposing regulations 

Dear Dr Lue, 

We are pleased to inform you that your manuscript entitled "Ustilago maydis Trf2 ensures genome stability by antagonizing Blm mediated telomere recombination: fine-tuning DNA repair factor activity at telomeres through opposing regulations" has been formally accepted for publication in PLOS Genetics! Your manuscript is now with our production department and you will be notified of the publication date in due course.

With kind regards,

Zsofia Freund

PLOS Genetics

On behalf of:
